# Novel Approach in Fracture Characterization of Soft Adhesive Materials Using Spiral Cracking Patterns

**DOI:** 10.3390/ma16237412

**Published:** 2023-11-29

**Authors:** Behzad Behnia, Matthew Lukaszewski

**Affiliations:** 1Department of Civil and Environmental Engineering, Clarkson University, Potsdam, NY 13699, USA; 2Department of Electrical and Computer Engineering, Clarkson University, Potsdam, NY 13699, USA; lukaszmp@clarkson.edu

**Keywords:** soft adhesives, asphalt materials, spiral cracking patterns, fracture characterization, acoustic emission, convolutional neural networks (CNNs)

## Abstract

A novel approach for the fracture characterization of soft adhesive materials using spiral cracking patterns is presented in this study. This research particularly focuses on hydrocarbon polymeric materials, such as asphalt binders. Ten different asphalt materials with distinct fracture characteristics were investigated. An innovative integrated experimental–computational framework coupling acoustic emissions (AE) approach in conjunction with a machine learning-based Digital Image Analysis (DIA) method was employed to precisely determine the crack geometry and characterize the material fracture behavior. Cylindrical-shaped samples (25 mm in diameter and 20 mm in height) bonded to a rigid substrate were employed as the testing specimens. A cooling rate of −1 °C/min was applied to produce the spiral cracks. Various image processing techniques and machine learning algorithms such as Convolutional Neural Networks (CNNs) and regression were utilized in the DIA to automatically analyze the spiral patterns. A new parameter, “Spiral Cracking Energy (*E_Spiral_*)”, was introduced to assess the fracture performance of soft adhesives. The compact tension (CT) test was conducted at −20 °C with a loading rate of 0.2 mm/min to determine the material’s fracture energy (G_f_). The embrittlement temperature (*T_EMB_*) of the material was measured by performing an AE test. This study explored the relationship between the spiral tightness parameter (“*b*”), *E_Spiral_*, G_f_, and *T_EMB_* of the material. The findings of this study showed a strong positive correlation between the *E_Spiral_* and fracture energies of the asphalt materials. Furthermore, the results indicated that both the spiral tightness parameter (“*b*”) and the embrittlement temperature (*T_EMB_*) were negatively correlated with the *E_Spiral_* and *G_f_* parameters.

## 1. Introduction

As a common phenomenon in materials, fractures can create various cracking patterns. It is especially prevalent in thin coatings under residual stresses, forming a network of cracking similar to the patterns observed in dried mud layers and old paintings, see Figure 1a. Some of the well-studied examples of fractures are straight-type cracks, also known as the mud (channeling) cracks, commonly observed in fragmented dried out fields, dried mud layers, coatings, and paintings. In addition to mud fracture patterns, which are the most common type, rare spiral-shaped cracking patterns have been observed in some soft adhesive materials, see Figure 1b.

Spiral cracks usually occur in a thin layer of soft materials coated on rigid substrates. Some distinct processes such as the drying, cooling, syneresis, or stretching of a substrate are attributed as the cause of the formation of this type of crack. Figure 2 schematically illustrates the mechanism of spiral crack development. The deformation mismatch between the coating layer and the substrate induces biaxial tensile stresses within the coating/substrate system, which will act as the driving force behind the crack propagation in the material. As the magnitude of the induced stresses increases, first a network of channeling cracks (i.e., mud cracks) appears hierarchically, dividing the layer into several polygonal sections. After the formation of mud cracks, the partial detachment of the material from the substrate (partial delamination) takes place within the polygonal sections. A delamination front begins from the edge of the polygonal cell and grows toward the center of the cell. While partial delamination occurs, the spiral crack starts at the border of the adhering region of the fragmentation from some initial imperfection or a weak interface and propagates along a spiral trajectory.

Dillard et al. investigated spiral cracks in LaRC™-TPI adhesive (a thermoplastic polyimide material). They found rapid cracking in the thin layer of this material when it was exposed to solvents, such as acetone, toluene, diglyme, and methyl ethyl ketone. They reported on the formation of mud cracking patterns dividing the coating into smaller sections, which is followed by the development of spiral cracks growing inward within each adhesive fragment [1,2]. Macnulty explored the occurrence of various types of cracks in biaxially stressed films of certain polymers containing phenylene, bonded to glass substrates. They reported on the formation of spiral cracks in addition to the regular cracking patterns in the material. They found that both spiral cracks accurately followed the logarithmic curve [3]. Hainsworth et al. investigated thin-film fractures in coating/substrate systems consisting of thin hard coatings deposited on a less stiff, hard substrate during depth-sensing indentation testing (e.g., nanoindentation testing). They conducted experiments on thin TiN and NbN monolayers and also on TiN/NbN and TiN/ZrN composites deposited onto hard steel or stainless steel substrates. They found that some of the cracks had a spiral morphology, which was not sensitive to the indentation loading rate [4]. Neda et al. explored the development of spiral cracks during the desiccation of thin films of precipitates on a substrate. The propagating stress front created by the fragments’ foldup was found to be the cause of this type of fracture. They showed that spiral cracks generally occur when the propagating speed of the stress front is proportional to the crack speed [5]. Volinsky et al. studied fractures in thin films caused by residual and/or externally applied stresses. Periodic spiral through-thickness cracks were reported in Mo/Si multilayers subjected to three-point bending in a vacuum at a high temperature [6]. Sendova et al. explored four distinctive cracking patterns including spiral cracks within thin silicate sol-gel films. Cracks geometry was found to be a function of the film thickness, curing time, and temperature. Their study concluded that spiral-shaped crack trajectories happen due to the local warping of the film caused by stress nonuniformity developed at the drying stage of the sol-gel material [7]. In another study, Meyer et al. investigated spiral cracking patterns in the Mo/Si multilayers (60 layers) with the thickness in the range of a nanometer deposited on Si substrates. Spiral cracks were observed in the material during uniaxial bending at high temperatures between 300 and 440 °C in a vacuum. The formation of spiral cracks, which was accompanied by Mo/Si multilayer debonding from the substrate, was linked to the combination of thermal, bending, and residual stresses [8]. Yonezu et al. investigated the initiation and propagation of spiral cracks in a thick layer of diamond-like carbon (DLC) deposited on a steel substrate during spherical indentation. They employed integrated acoustic emission (AE) and corrosion potential fluctuation (CPF) testing methods to evaluate the cracking process. Their experiments showed that spiral cracks can develop only within a narrow range of the maximum indentation force. In case the indentation force is below that range, no fracture happens, and if it is above the range, the ring cracks are formed [9]. Marthelot et al. explored the spiral cracking phenomenon and reported that this type of crack can occur below the standard critical tensile load required for mud cracks. They also demonstrated that spiral cracks choose a robust interaction length scale, which could be 30 times the film thickness [10]. Monev investigated the spiral cracking phenomenon in nickel coatings electrodeposited in an acidic media containing hydrogenation-enhancing additives. The results demonstrated that the type of the additives affects the shape of the cracks as well as the tendency of the nickel layers to form cracks [11]. Wu et al. reported on the formation of perfect Archimedes spiral cracks in a colloid film with several hundred nanometers thickness deposited on a glass substrate [12]. Matsuda et al. studied the occurrence of relatively large-sized spiral cracks (with diameters greater than 0.4 mm) on the surface of melt-grown poly(L-lactic acid) (PLLA) spherulites. The spiral patterns developed in the material due to the thermal shrinkage occurring upon cooling after crystallization. The spiral pitch was found to be positively correlated with the thickness of the spherulite, meaning that the increase in the thickness of the spherulite led to an increase in the spiral pitch [13]. Ma et al. studied spiral cracks in drying suspensions of Escherichia coli (E. coli) with different swimming behaviors. They used 2.5 µL of bacterial suspensions deposited on glass substrates for drying. The spiral cracks were observed in the film of the mutant E. coli with tumbling motions, while the circular cracks were found in the consolidating film of the wild-type E. coli. It was demonstrated that the spiral cracks occur due to film delamination that is caused by the strong bending moment [14], see Figure 3. Behnia et al. reported on the formation of spiral fracture patterns inside a thin layer of hydrocarbon polymeric materials such as asphalt binders and further investigated these patterns [15]. They also explored the potential use of spiral cracking patterns for the fracture characterization of asphalt materials at different oxidative aging levels. They reported that the spiral tightness parameter was found to be sensitive to both the oxidative aging level and the performance grade of the asphalt materials [16]. The logarithmic spiral function was found to be the best fit to mathematically represent 3D spirals, see Equation (1), where “A” is the apparent length scale, “*b*” is the spiral tightness parameter, $D_{0}$ is the initial spiral crack depth, θ is the angle from the *x*-axis, and $\theta_{f}$ is the final angle corresponding to the outmost point of the spiral in the sample:(1)$$\vec{P\left(r,\theta ,Z\right)}=Ae^{b\theta }cos\left(\theta \right)\vec{i}+Ae^{b\theta }sin\left(\theta \right)\vec{j}+\frac{\theta }{{2\theta }_{f}}D_{0}\;\vec{k}$$

This study presents a transformative and radically different approach using spiral cracking patterns as a powerful diagnostic tool to obtain valuable information about the fracture characteristics of soft adhesives, particularly hydrocarbon polymeric materials. The accurate fracture characterization of soft adhesive materials has remained a challenging task. The implementation of conventional testing methods such as the compact tension (C(T)) test (to evaluate the fracture characteristics) and the Peel test (to measure the adhesive strength) of such materials is quite challenging. Due to the soft nature of these materials, large creep deformations usually occur in the material during the experiment. Moreover, another concerning issue is that the Linear Elastic Fracture Mechanics-based (LEFMs) theories are not suitable to describe the fracture of soft and highly stretchable materials. Research studies have demonstrated that due to large deformations during the fracture process, the stress field near the crack tip in soft materials is significantly different from that used for LEFMs (LEFMs is based on the assumption of infinitesimal deformations) [17].

## 2. Materials

The present work utilizes hydrocarbon polymeric materials such as asphalt binders to investigate the methodology. It is important to highlight that while the methodology is demonstrated using hydrocarbon polymers, it can be applied to other soft adhesives with minor adjustments in the sample geometry and substrate material selection. Ten different types of asphalt materials (here referred to as *AB1*, *AB2*,…, *AB10*) with different fracture properties and the following performance grades (PG) were utilized in this work: *PG58-10*, *PG58-16*, *PG58-22*, *PG58-28*, *PG58-34*, *PG64-10*, *PG64-16*, *PG64-22*, *PG64-28*, and the *Styrene-butadiene-styrene (SBS)-modified PG64-22*. The modification of asphalt binders with SBS is a common technique used to enhance the mechanical properties of asphalt materials. The conventional notation for the performance grading of asphalt materials is *PG XX-YY*, where “*XX*” and “*YY*” represent the PG high and low temperatures, respectively. The PG high temperature refers to the average maximum temperature (°C) that an asphalt road sustains over a seven-day period. On the other hand, the PG low temperature signifies the minimum temperature (°C) that the asphalt road is likely to encounter throughout its service life [18].

## 3. Spiral Cracking Experiment

Cylindrical-shaped specimens bonded to a rigid substrate were used as the testing configuration for semi-solid soft adhesives, see Figure 4a. Depending on the type of adhesive material being tested, the diameter (D) and thickness (h) of the specimen as well as the type of substrate material (such as glass, aluminum, etc.) can be different and they should be carefully selected to ensure there are no delamination and no mud cracks within the spiral specimen. For hydrocarbon polymeric materials, various geometries with different thicknesses (ranging from 1 mm to 30 mm) were investigated. The results demonstrated that the cylindrical specimens with D = 25 mm and h = 20 mm bonded to an aluminum substrate showed the most promising results (i.e., no debonding and no mud cracks). The lack of mud cracks in the proposed testing configuration is an advantage that minimizes the variation in the spiral results, thereby enhancing repeatability. Through this investigation, it became evident to the authors that the presence of mud cracks within the sample introduces complexity to the analysis, significantly increasing variations in the results (i.e., reduces repeatability). The fracture-originated acoustic emissions (AE) signals are used for determining the overall 3D geometry of a spiral crack as well as finding the characteristic parameters of the spirals, such as the spiral tightness parameter. In the case of using other sample geometries such as a prismatic shape, the formation of spiral cracks is accompanied by the occurrence of mud cracks in the specimen. The length, formation location, and orientation of mud cracks exhibit variability from one sample to another, introducing complexity to the analysis of the AE signals. The recorded AE signals in the samples with mud cracks can be categorized into two parts: One part is the AE signals originating from the spiral crack (this part is used in the analysis). Another part is the AE signals stemming from the mud cracks. Variations in the geometric characteristics of the mud cracks result in notable differences in the number of AE signals, their energy content, amplitude, frequency content, etc. Consequently, this variability contributes to an increase in the Coefficient of Variation (CoV%) of the results and makes the interpretation of the spiral cracking results complicated.

Spiral cracks are produced in the lab under a controlled condition. Depending on the type of material, they could be thermally induced, solvent-induced, or caused by drying. In the case of hydrocarbon polymeric materials, spiral cracks are thermally induced and produced by cooling the sample. In the preliminary study, various cooling rates including 0.3, 0.5, 1, 1.5, and 2 °C/min were investigated. The results showed that cooling rates higher than 1 °C/min resulted in sample delamination from the substrate before the spiral crack even gets a chance to develop. Thus, the average cooling rate of 1 °C/min was applied for testing the asphalt materials. The asphalt samples were prepared by pouring heated material into a cylindrical-shaped silicon mold mounted on the substrate. The samples were allowed to reach room temperature before the testing. To conduct the experiment, the sample was cooled down from 0 to −50 °C at the average rate of 1 °C/min. We selected −50 °C to make sure the spiral crack was fully developed inside the sample. The results showed that the formation of a spiral crack in the asphalt samples subjected to a 1 °C/min cooling rate was complete at temperatures ranging from −40 °C to −50 °C for various types of asphalt. As the temperature reduces, differential thermal contraction between the rigid substrate and asphalt creates thermally induced stresses within the material, which eventually leads to the formation of an inward-growing spiral crack nucleating from some initial imperfection or a weak interface near the edge of the specimen.

The results showed a gradual reduction in the spiral crack depth (penetration depth) from almost 50% of the sample thickness (h) at the edge to almost zero at the center of the specimen, see Figure 4b. This phenomenon could be linked to a gradual reduction in the amount of stored strain energy as the crack spirals from the edge toward the center of the sample. During the fracture process, the stored strain energy in the specimen is consumed for the creation of new fractured surfaces. At the beginning of the fracture process, the stored strain energy in the specimen is at the highest level, leading to the nucleation of the spiral crack at the interface, with a maximum penetration depth, $D_{0}$. As the spiral crack grows, the strain energy is gradually diminished to create new fractured surfaces. As a result, the crack penetration depth continuously decreases until it reaches almost zero at the center of the sample. The careful inspection of a fully grown 3D spiral showed that the gradual change in the spiral crack depth is almost linear. Figure 4c schematically shows the hypothetical unwrapped shape of a 3D spiral crack in the form of a triangle.

## 4. Integrated Experimental and Computational Framework

An overview of the integrated experimental and computational framework consisting of the multi-sensor *AE* method coupled with the Digital Image Analysis (*DIA*) approach to evaluate the fracture characteristics of soft adhesive materials is illustrated in Figure 5 (the technical details of these methods are provided in Section 4.1 and Section 4.2). In this approach, the total area of the spiral-shaped fractured surfaces (*A_Fracture_*) inside the specimen is calculated through the application of the coupled *AE-DIA* approach, where the total length of the spiral crack ($L_{Spiral}$) is measured using *DIA* and the initial depth of the spiral crack $D_{0}$ is determined using the *AE* source location. For the *DIA* analysis, a novel machine learning-based image processing framework is applied. The AE energies of individual events (i.e., the *AE* event is a rapid physical change, such as microcracks appearing as an acoustic signal) will be added up to measure the total amount of released *AE* energy due to the formation of a spiral crack within the sample. A new parameter (index) called the spiral cracking energy (*E_Spiral_*) is introduced, which is defined as the amount of released *AE* energy per unit of the newly formed fracture surface area of the spiral cracks. It should be noted that this fracture index is not the same as the fracture energy of the material due to the fact that the measured *AE* energy is not equal to the strain energy released during crack propagation. During the spiral cracking process, part of the strain energy released in the specimen is used to create new fractured surfaces, and the rest is released as transient elastic waves, which can be picked up by *AE* sensors (a portion of transient waves can be dissipated by attenuation before reaching *AE* sensors). The former is related to the fracture energy and the latter is captured by the *AE* method. The *E_Spiral_* index quantifies the fracture resistance of soft adhesive materials using their *AE* activities during the spiral cracking. The unit of *E_Spiral_* is *V*^2^.μs/mm^2^ and it can be calculated by dividing the total released *AE* energies of the fracture-induced signals by the total fractured surface area within the sample, Equation (2), where *N* is the total number of recorded *AE* signals, *V_i_(t)* is the voltage of the *i*th recorded signal in volts, and $A_{Fractured}$ is the total surface area of the fractured faces measured from the *AE-DIA* analysis.
(2)$$E_{spiral}=\frac{\sum_{i=1}^{N}\int_{0}^{t}V_{i}^{2}\left(t\right)dt}{A_{Fractured}}$$

### 4.1. Multi-Sensor Acoustic Emission Testing

The multi-sensor acoustic emission (*AE*) technique was employed to continuously monitor the acoustic activities of the specimen during the course of the spiral cracking experiment and also to measure the initial depth of the spiral crack, $D_{0}$, see Figure 6. To prepare the cylindrical asphalt samples with a diameter of 25 mm and height of 20 mm, hot liquified asphalt at 135 °C was poured into a cylindrical silicon rubber mold mounted securely on an aluminum plate. To facilitate the sample demolding process, the silicon model was covered with Teflon tape. To ensure proper bonding between the asphalt and the substrate, the aluminum plate was heated to 135 °C before pouring the binder. The sample was let to cool down to room temperature and then placed in the freezer (set at 0 °C) for a few minutes to make the demolding process easy. A spiral crack developed inside the material by cooling the sample from room temperature to −50 °C at an average cooling rate of −1 °C/min.

*AE* sensors with a relatively flat response over the target frequency range capable of working properly in the target range of the test temperatures were utilized. For the asphalt materials, broadband *AE* sensors with flat responses in the frequency ranging 20 kHZ–1 MHz were used. To minimize the extraneous noise (i.e., separating genuine fracture-originated *AE* signals from noise), the signals were pre-amplified to 20 dB using broadband pre-amplifiers. The signals were then further amplified by 21 dB (for a total of 41 dB) and filtered using low-pass (LPF) and high-pass (HPF) filters of 500 kHz and 20 kHz, respectively, with the Fracture Wave Detector (FWD) signal condition unit. The signals were digitized using a 16-bit analog-to-digital converter and a sampling frequency of 1 MHz and a length of 2048 points per channel per acquisition trigger. At the postprocessing stage, all the AE signals with energy lower than 4 *V*^2^-μs were filtered out. Moreover, the standard pencil-lead break (PLB) test was performed routinely before conducting the experiments in order to calibrate the *AE* system as well as the *AE* sensors and to make sure the variation within the *AE* channels was negligible.

As the spiral crack propagates inward, new fractured surfaces are formed, which are accompanied by the release of stored strain energy in the form of transient mechanical stress waves inside the specimen. The *AE* piezoelectric sensors mounted on the surface of the specimen as well as on the substrate adjacent to the specimen will continuously monitor and detect these mechanical waves and convert them into *AE* signals. The *AE* signals carry valuable information about the spiral fracture process occurring in the material. Therefore, the *AE* signals were recorded, carefully analyzed, and the key features of the signals such as the signal amplitude, energy, frequency, hit counts, arrival time, and duration were extracted.

In addition, the *AE* source location technique was applied to measure the starting penetration depth of the spiral crack (*D_0_*). An integrated approach combining the two-step *Akaike Information Criterion (AIC)* method [19] and Simplex algorithm (also known as the Nelder–Mead algorithm) was implemented. The *AIC* approach was used for the precise automatic determination of the time of arrival (*ToA*) of the *AE* signals. The autoregression-based *AIC* function divides the *AE* signal, $\left\{X_{1},X_{2},\ldots ,X_{N}\right\}$, into two vectors at the time k: $\left\{X_{1},X_{2},\ldots ,X_{k}\right\}$ and $\left\{X_{k+1},X_{k+2},\ldots ,X_{N}\right\}$. The method then compares the signal variance of the prediction errors before and after the time k in a predetermined time window. The *AIC* value at point i=k is calculated using Equation (3), where N is the number of amplitudes of a digitized *AE* wave; $X_{i}$ is an amplitude of a signal (i=1,2,..,N); $var(X\left[1,k\right])$ is the variance of X between $X_{1}$ and $X_{k}$; and $var(X\left[k+1,N\right])$ is the variance of X between $X_{k+1}$ and $X_{N}$. The *ToA* of an *AE* signal is the time at which the *AIC* function becomes a global minimum. In the two-step *AIC* process, in the first step, the global minimum of the *AIC* function is employed to obtain the first estimation of the *ToA*. In the second step, the time interval is narrowed down and focused on the neighborhood of the first *ToA* estimation. The final value of the *ToA* of the signal is computed using the global minimum of the recalculated *AIC* function.
(3)$${AIC}_{k}=k.log\left\{var\left(X\left[1,k\right]\right)\right\}+\left(N-k-1\right).log\left\{var\left(X\left[k+1,N\right]\right)\right\}$$

After calculating the *ToA*s of the *AE* signals, the Simplex algorithm was used for the source location. In this method, for any point in the medium, an error (E) was computed by comparing the calculated and observed arrival times. As such, an error space is created, in which each point is an error associated with a point in the specimen. The point with the minimum error is the event location. Equation (4) is used to calculate the location error using the least squares method ($L_{2}$
*norm*), where E is the least squares error ($L_{2}$
*norm error*); $t_{i}$ is the *ToA* obtained from the *AIC* picker; ${tt}_{i}$ is the travel time from the location of interest to the ith sensor; n is number of *AE* sensors; m is number of equations; and q is the degree of freedom. The error space is created by computing E for every point in the medium and the point with the minimal error is localized as the source location of the *AE* event.
(4)$$E=\sqrt{\frac{\sum {\left[\left(t_{i}-\frac{\sum t_{i}}{n}\right)-\left({tt}_{i}-\frac{\sum tt_{i}}{n}\right)\right]}^{2}}{m-q}}$$

The accuracy of the source localization was assessed using the *Euclidean Distance Error (EDE),* in which the *PLB* procedure (*Hsu–Nielsen* source) is used to artificially generate *AE* signals at various locations of the sample. The source of the *AE* activities was located and compared with the actual locations. The *EDE* was measured by calculating the distance between the actual source location and the estimated source location. The results showed around 92% accuracy of the source location approach in the spiral specimens.

In addition to determining the crack’s initial penetration depth, the *AE* source location was also implemented to measure the 3D shape and size of the fracture process zone (*FPZ*) ahead of the spiral crack tip. The results showed that unfortunately the source location did not have a sufficient spatial resolution to map out the accurate 3D geometry of the *FPZ* in the spiral specimens. Figure 7 illustrates a typical map of *AE* activities within the profile of the spiral specimen obtained through an *AE* source location analysis. The gray dashed lines represent the boundaries of the spiral specimen (25 mm in diameter and 20 mm in height), while the red dots indicate the locations of the *AE* events that occurred within the sample. The results of the *AE* source location analysis demonstrate that the initial depth of the spiral crack (*D*_0_) can be accurately measured, see Figure 7. However, due to the limitations in the spatial resolution of the *AE* source location, it was not possible to precisely detect the exact location of the crack front and measure its depth as it spirals toward the sample center.

The *AE* test was also used to measure the embrittlement temperature (*T_EMB_*) of the material. The *T_EMB_* is considered as the starting point of thermally induced damage in the material. By analyzing the *AE* signals in conjunction with the recorded test temperatures, the temperature corresponding to the occurrence of the *AE* event with the first peak energy level was identified. This temperature is used as the *T_EMB_* of the material. 

### 4.2. Machine Learning-Based Framework for Digital Image Analysis (DIA)

The pipeline of the framework used for the automated *DIA* of the spiral cracking patterns is illustrated in Figure 8. It consists of various image processing techniques along with the *Convolutional Neural Networks* (*CNNs*) machine learning approach. In the first step, image skeletonization was applied, in which a spiral cracking image was reduced to a one-pixel-thick skeleton of a crack path and the output image was converted into grayscale. Skeletonization was performed to accelerate the analysis by providing a light skeleton for image processing instead of an otherwise computationally expensive analysis on the original image. After skeletonization, the Gaussian blurring filter and Sobel method were applied to remove the inhomogeneous image background illuminations as well as for the edge detection of the cracking patterns, respectively. This was followed by the application of the Hough transform method to detect the spiral shapes in the image. This approach is capable of detecting shapes in images even if those shapes are slightly broken or distorted [20].

The *CNN* algorithm was implemented for noise reduction in the skeletonized images. This algorithm is ideal for image processing tasks as it significantly reduces the number of required weights for neurons in the model by using tiling regions, each with the same shared weights. The *CNN* autoencoder compresses spiral cracking images in a series of convolutions and then reverses the process during the decoding step. The hidden layers of the *CNN* perform convolutions generating a feature map, which contributes to the input of the next layer. Each convolutional layer is followed by a Rectified Linear Unit (*ReLU*) layer. A regular feedforward neural network consisting of a couple of fully connected layers (*+ReLUs*) is added at the beginning of the stack and the final layer outputs the prediction using the *Softmax* activation function. To properly train the *CNN* model in addition to real spiral crack images, a large number of synthetic images of spiral patterns with various shapes and sizes were generated. Random artificial noise was added to both the real and synthesized spiral images. A combination of real and synthesized images with and without artificial noise was used for training the *CNN*. Once the noise was removed from the images, the segmentation technique was performed to separate and extract the spiral crack path from the original and was followed by measuring the total length of the spiral crack (*L_Sprial_*). Finally, a regression analysis was employed to determine the spiral parameters: “*A*” and “*b*”.

## 5. Results and Discussion

Table 1 summarizes the spiral cracking parameters from the *AE-DIA* approach, the fracture energy (*G_f_*) values obtained from compact tension (*C(T)*) test, and the embrittlement temperature (*T_EMB_*) from the AE test. Each data point in Table 1 is the average of six to eight testing replicates. Performing a *C(T)* test for semi-solid adhesives is a challenging task due to the soft nature of these materials. In this study, various temperatures ranging from 0 °C to −30 °C and the loading rates (0.1-1 mm/min) were carefully explored. It was observed that for the asphalt binders, the *C(T)* tests performed at −20 °C with a loading rate of 0.2 mm/min were satisfactory. The *C(T)* test was performed in accordance with ASTM E399-05 to determine the fracture energy of the asphalt binders [21]. To prepare the *C(T)* samples, hot liquified asphalt at 135 °C was poured into a silicon rubber mold and the samples were left to cool down. After reaching room temperature, the samples were placed in the freezer for 5 min to make the demolding process easy. The *C(T)* samples were conditioned for two hours at −20 °C before conducting the experiment. The fracture energy (*G_f_*) of the material was computed by dividing the calculated area under the load–CMOD curve by the fractured surface area. Some fracture energy data are missing in Table 1 primarily because the authors encountered sample failure during the *C(T)* tests, preventing them from successfully conducting the experiments for those materials.

The results showed that the *E_Spiral_* values were positively correlated with the fracture energies of the asphalt materials, meaning that the average amount of released *AE* energy per unit area of spiral cracks is higher for materials with higher resistance against cracking. A comparison of the *E_Spiral_* and *G_f_* with the spiral tightness, “*b*”, values demonstrated a different trend in which both *E_Spiral_* and *G_f_* were negatively correlated with “*b*”. This could be explained through understanding the mechanism behind the formation and propagation of spiral cracks. It is hypothesized that a spiral crack front selects its trajectory in a direction where it could maximize the stored strain energy release rate [22]. As such, spiral cracks have a constant pitch angle, meaning that the orientation of the crack front is constantly bending away from the instantaneous propagation direction. The spiral pitch angle is lower in high fracture-resistant materials, because it is more difficult for the crack front to bend away from its instantaneous propagation direction in high fracture-resistant materials. A lower spiral pitch angle (*φ*) results in a lower spiral tightness parameter “*b*” (*b* = tan(*φ*)). The findings of this study showed the dependency of the *G_f_* of asphalt to its PG temperatures, particularly to the PG low temperature (*PGLT*) of the material, where binders with a lower *PGLT* exhibited higher fracture energies [15,16]. Additionally, it was observed that the *PGLT* of the binder has a significant effect on its “*b*” value. Generally, the lower the *PGLT* of a binder, the higher its low-temperature cracking resistance (higher fracture energy). As a result, asphalt materials with a lower *PGLT* tend to have lower “*b*” values. Furthermore, it is noteworthy that the asphalt modified with *SBS* demonstrates the highest fracture energy and a relatively low spiral tightness parameter. This outcome can be attributed to the influence of the *SBS* polymer modification to enhance the resistance of the material against cracking.

The *AE* results suggest a strong correlation between the *AE* embrittlement temperature (*T_EMB_*), spiral tightness parameter, and fracture energy of the asphalt materials. The observations reveal a positive correlation between the *T_EMB_* of the binders and the “*b*” value, while a negative correlation exists between the *T_EMB_* and the fracture energy of the material. A summary of the observed correlations between the spiral tightness (b), spiral cracking energy (*E_Spiral_*), embrittlement temperature (*T_EMB_*), and fracture energy (*G_f_*) for the hydrocarbon polymeric materials is presented in Table 2.

Typical plots of the *AE* results including the cumulative *AE* hit counts and *AE* energy versus temperature are presented in Figure 9. The *AE* response of the material during spiral cracking showed four distinct regions: region#1 (pre-cracking region), region#2 (transition region from quasi-brittle to brittle state), region#3 (stable cracking region), and region# 4 (fully cracked region). The magnitude of the emitted *AE* energies is highest when the spiral crack starts to propagate (at the start of region#2). As the crack continues to grow, the magnitude of the emitted *AE* energies gradually tapers off until it reaches almost zero at the end of region#4 when the spiral crack is fully developed. This can be linked to a gradual reduction in the stored strain energy (as the driving force behind crack propagation) when the crack front advances from the edge toward the center of the sample. Figure 4 schematically shows the 3D spiral cracking pattern with the gradual reduction in crack penetration depths. During the fracture process, the stored strain energy in the specimen is consumed for the creation of new fractured surfaces. At the end of region#1 (the pre-cracking region), the stored strain energy in the specimen is at the highest level, leading to the nucleation of the spiral crack at the interface with a maximum penetration depth, $D_{0}$. As the crack grows, the strain energy is gradually diminished to create new fractured surfaces. As a result, the crack penetration depth continuously decreases until it reaches almost zero at the center of the sample. A visual inspection of the fully grown 3D spiral showed that the gradual change in the spiral crack depth is almost linear. Figure 4 demonstrates the hypothetical unwrapped shape of the 3D spiral crack in the form of a triangle.

A further analysis of the *AE* signals was performed to investigate other important *AE* parameters, such as the signal duration time, signal rise time (*RT*), frequency content, rise angle (*RA* = *Signal rise time/Signal peak amplitude*), and average frequency (*AF = AE hit counts/Signal duration time*). An interesting observation was that the *AE* signals recorded at the beginning of the spiral crack formation had a low rise time, high amplitude values, and a high frequency (low *RA* and high *AF*) (this type of signal is usually observed in fracture Mode I), while the signals recorded at lower temperatures exhibited a longer duration time, longer rise times, a low amplitude, and a lower frequency (high *RA* and low *AF*) (these types of signals are typically observed in fracture *Mode II*), see Figure 9c and Figure 10. These findings suggest that during the course of the spiral cracking process, the fracture mode changes from *Mode I* to *Mode II*. It was observed that the change in the *AF* and *RA* values mostly occurred at temperatures near the glass transition temperature (*T_g_*) of the material, where the material behavior gradually changes from quasi-brittle to a brittle state (the *T_g_* values for the asphalt materials utilized in the present study were in the range of −25 °C to −30 °C). As a result, it is hypothesized that a change in the fracture mode of spiral cracks happens at temperatures close to the glass transition temperature of the material (within the transition region (region#2)). A more in-depth investigation is required to further explore this hypothesis.

## 6. Conclusions

The present work investigates a novel approach for the fracture characterization of soft adhesives using spiral cracking patterns. Ten different hydrocarbon polymeric materials with various fracture characteristics were utilized in this study. Cylindrical-shaped samples (with a 25 mm diameter and 20 mm height) bonded to a rigid substrate (aluminum plate) were used as the testing specimens for the hydrocarbon polymeric materials. A spiral crack formed inside the specimen by cooling it from room temperature to −50 °C at an average rate of −1 °C/min. An integrated experimental–computational framework coupling the multi-sensor *AE* and *DIA* approaches was employed to determine the spiral cracking parameters, such as the spiral tightness parameter. An efficient image processing framework using various image processing and machine learning techniques such as *CNNs*, skeletonization, segmentation, and regression were used in the *DIA* for the automatic analysis of the spiral pattern images. A new parameter called the “*Spiral Cracking Energy* (*E_Spiral_*)” was introduced to evaluate the fracture performance of the soft adhesives. 

In addition to the spiral cracking parameters, some other critical parameters used in the low-temperature cracking evaluation of the hydrocarbon polymeric materials such as the fracture energy and the embrittlement temperature were measured. The compact tension (CT) test was conducted at −20 °C with a loading rate of 0.2 mm/min to determine the fracture energy (*G_f_*) of the material. The embrittlement temperature (*T_EMB_*) of the material was determined by performing an AE test. A summary of the observed correlations between the spiral tightness (*b*), spiral cracking energy (*E_Spiral_*), embrittlement temperature (*T_EMB_*), and fracture energy (*G_f_*) are presented in the following:Exploring the relationship between the *E_Spiral_* and *G_f_* values of the hydrocarbon polymeric materials showed a strong positive correlation between these two parameters, where an increase in one leads to an increase in the other one, and vice versa.Additionally, the results indicated that the spiral tightness parameter (“*b*”) was negatively correlated with both the *E_Spiral_* and *G_f_* parameters for the hydrocarbon polymeric materials, meaning that the increase in the spiral tightness parameter is associated with a decrease in the *E_Spiral_* and *G_f_* parameters, and vice versa.Investigating the relationship between the *T_EMB_* and spiral tightness parameter revealed a positive correlation between the *T_EMB_* and the “*b*” value in the hydrocarbon polymeric materials.The experimental results also showed a negative correlation between the *E_Spiral_* and *G_f_* parameters and the *T_EMB_* of the hydrocarbon polymeric materials.

The characteristic parameters of the AE signals such as the duration time, rise time (*RT*), frequency content, rise angle (*RA*), and average frequency (*AF*) were analyzed. It was observed that at the beginning of the spiral cracking process, the signals exhibited a low *RA* and high *AF* (usually observed in fracture Mode I). On the other hand, later, at lower temperatures (lower than the glass transition temperature), the recorded AE signals were mostly of a high *RA* and low *AF* (typically observed in fracture Mode II). It is hypothesized that a change in the fracture mode may happen at temperatures near the glass transition temperature (*T_g_*) of the material, at which the material behavior changes from quasi-brittle to brittle.

Given the current challenges in carrying out conventional fracture tests for soft adhesives, the findings of this limited study suggest that the use of spiral cracking patterns could be considered as a viable alternative for assessing the fracture characteristics of such materials and laying the groundwork for future advancements in this field. Particularly in the field of hydrocarbon polymeric materials, numerous studies have been directed toward the low-temperature cracking characterization of asphalt materials [23,24,25,26,27,28,29]. It is recommended that for future studies, the spiral cracking results are compared against those from the existing testing methods, such as the Asphalt Binder Cracking Device (ABCD), Bending Beam Rheometer (BBR), etc. Such a comparative analysis aims to further explore the correlations and establish relationships between different testing approaches, paving the way for enhanced insights into the fracture performance of asphalt materials.

## Figures and Tables

**Figure 1 materials-16-07412-f001:**
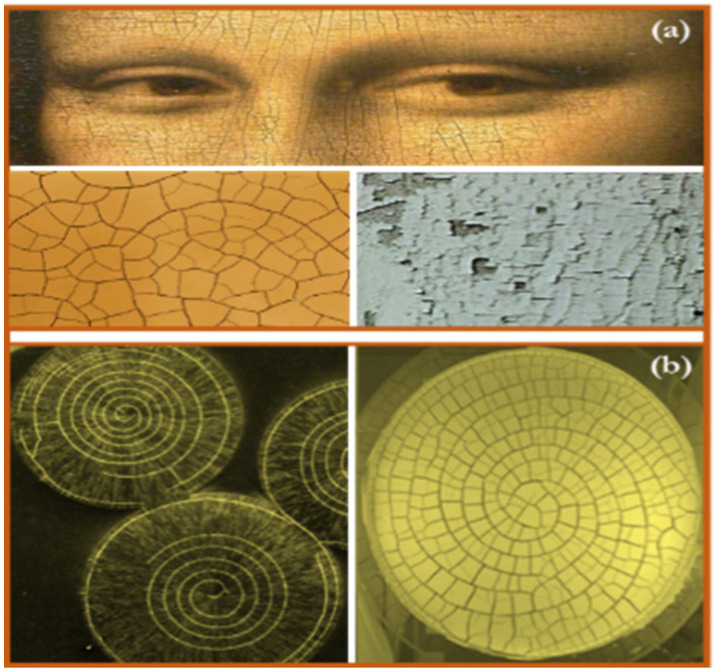
Various fracture patterns in materials. (**a**) Mud cracking patterns. (**b**) Spiral fracture patterns.

**Figure 2 materials-16-07412-f002:**
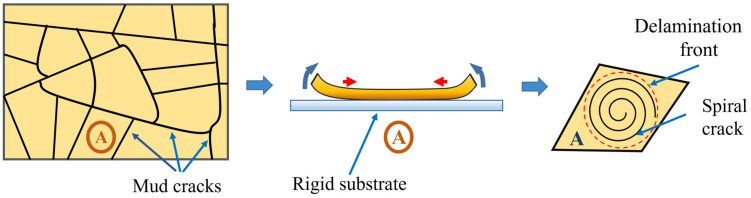
Mechanism of formation of spiral cracks in coating/substrate systems.

**Figure 3 materials-16-07412-f003:**
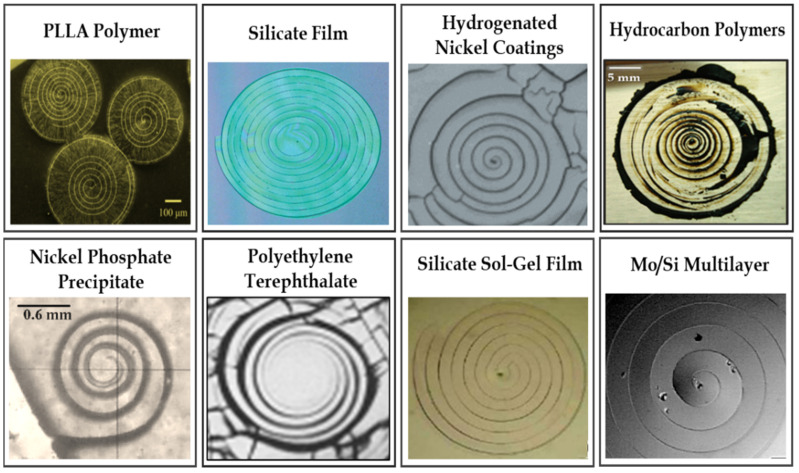
Spiral cracks in different materials [3,5,6,7,10,11,13,14].

**Figure 4 materials-16-07412-f004:**
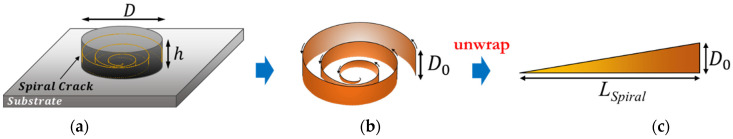
(**a**) Testing configurations for semi-solid soft adhesives. (**b**) Three-dimensional spiral crack developed in the sample. (**c**) Unwrapped spiral crack in the form of a triangle.

**Figure 5 materials-16-07412-f005:**
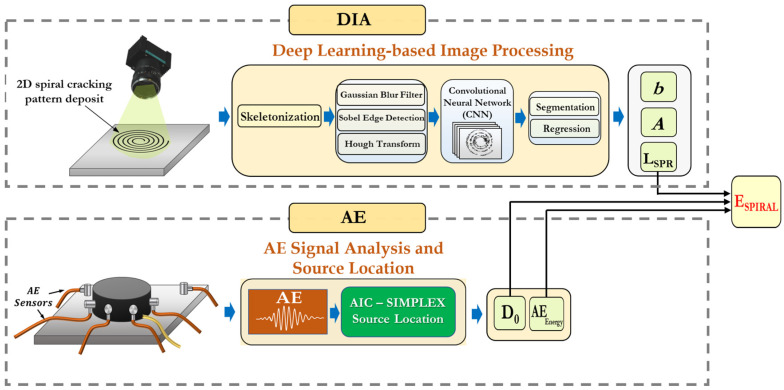
Integrated *EA-DIA* approach to assess fracture characteristics of soft adhesives.

**Figure 6 materials-16-07412-f006:**
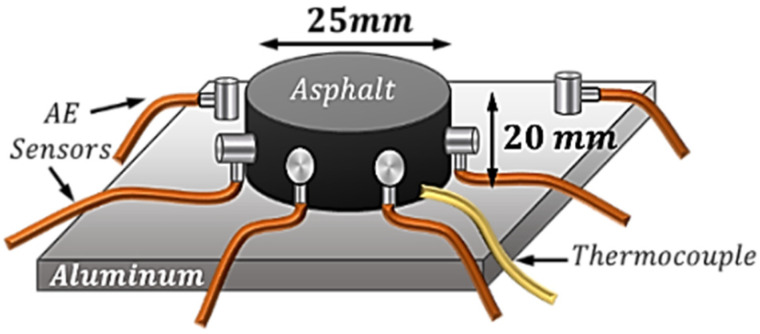
Testing specimens used for hydrocarbon polymeric materials.

**Figure 7 materials-16-07412-f007:**
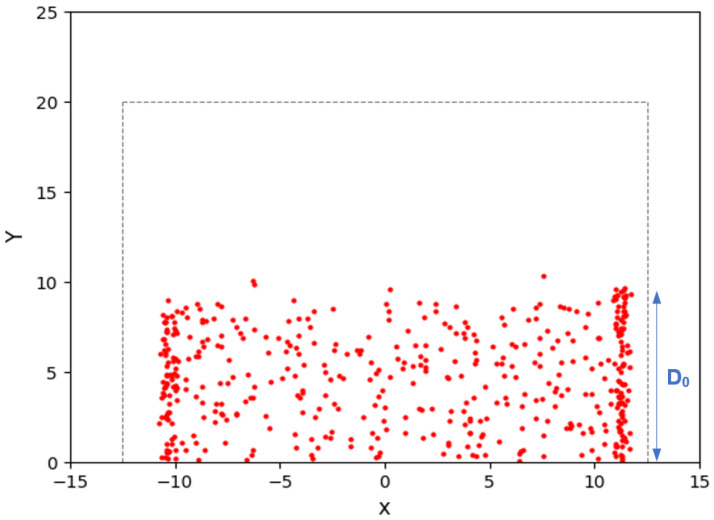
Map of AE events activities (red dots) within spiral cracking sample.

**Figure 8 materials-16-07412-f008:**
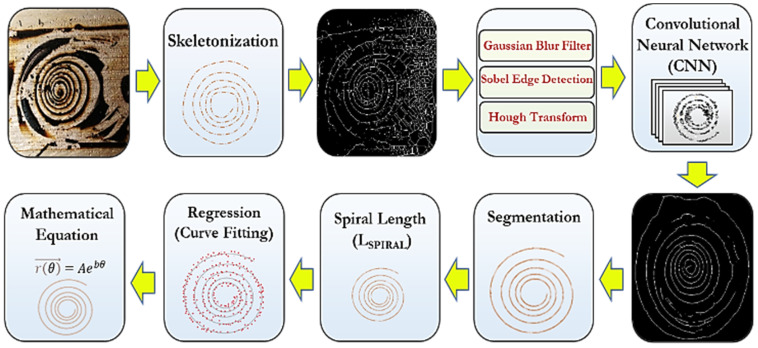
Pipeline of the ML-based DIA framework.

**Figure 9 materials-16-07412-f009:**
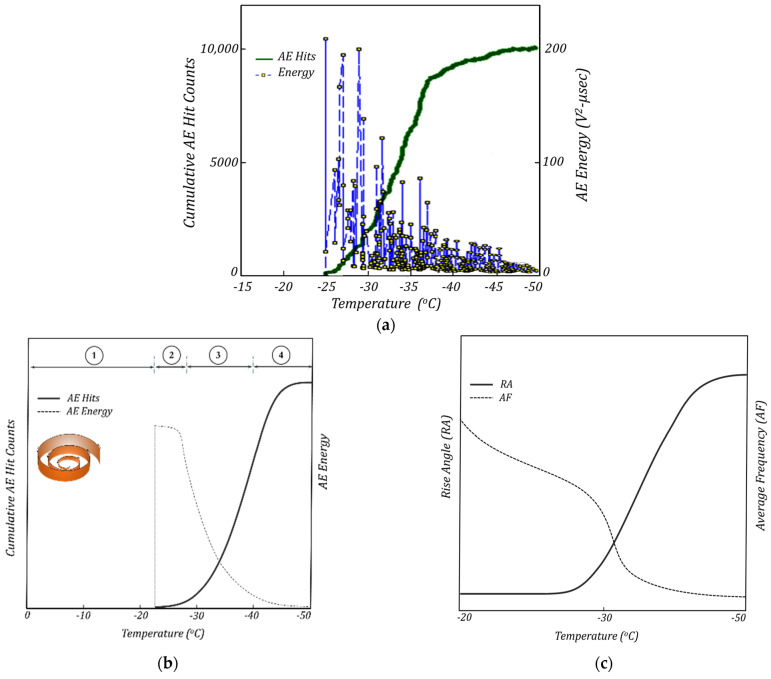
(**a**) Typical plot of cumulative AE hit counts and AE events energy versus temperature. (**b**) Typical envelope locus of AE hit counts and AE energies vs. temperature. (**c**) AE rise angle and average frequency vs. temperature.

**Figure 10 materials-16-07412-f010:**
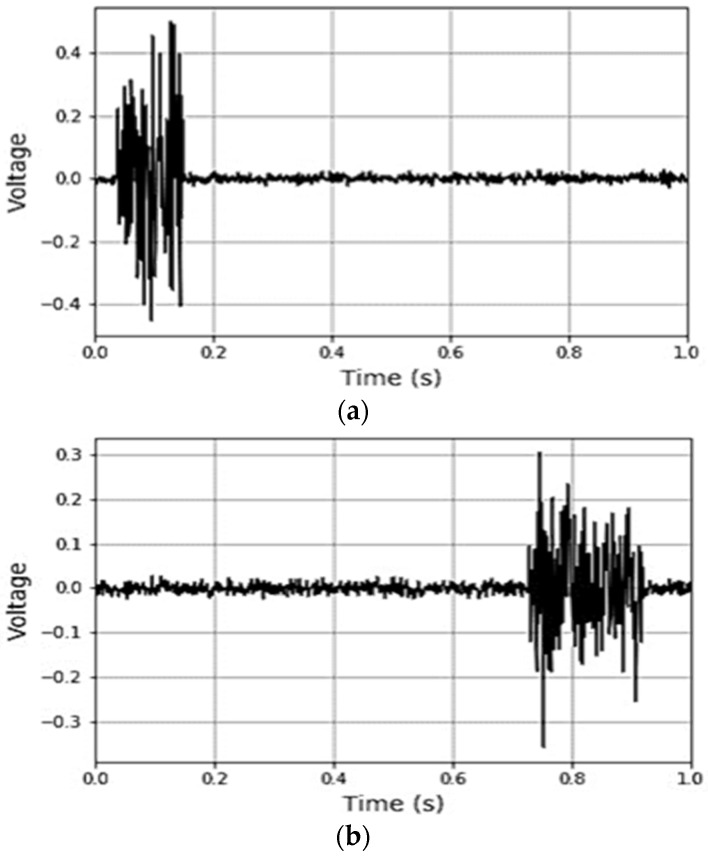
(**a**) Typical *AE* signal at temperatures above the glass transition temperature of the material: low rise time, high amplitude values, and high frequency (low *RA* and high *AF*). (**b**) Typical *AE* signal at temperatures below the glass transition temperature of the material: longer waveforms, with longer rise times, low amplitude, and lower frequency (high *RA* and low *AF*).

**Table 1 materials-16-07412-t001:** Experimental results for ten different asphalt materials.

Material ID	Performance Grade (PG)	*Φ* (deg)	*b*	CoV (%)	*E_Spiral _* (*v*^2^.μs/mm^2^)	CoV (%)	*AE**T_EMB_* (°C)	CoV (%)	*G_f_* (J/m^2^)	CoV (%)
AB-1	PG 58-10	3.907	0.0683	9.2	147	11.5	−15.2	12.6	-	-
AB-2	PG 58-16	2.674	0.0467	11.6	435	15.9	−29.6	6.5	2.42	18.3
AB-3	PG 58-22	2.742	0.0479	7.3	411	8.5	−31.9	9.1	8.27	13.9
AB-4	PG 58-28	2.411	0.0421	13.8	761	10.1	−34.3	13.3	13.09	16.2
AB-5	PG 58-34	1.919	0.0335	10.5	1092	9.7	−39.8	10.5	34.11	21.7
AB-6	PG 64-10	3.388	0.0592	5.7	185	14.9	−17.4	8.6	-	-
AB-7	PG 64-16	2.594	0.0453	9.3	335	10.2	−28.0	12.1	-	-
AB-8	PG 64-22	2.525	0.0441	12.2	455	8.4	−31.1	7.2	3.14	15.5
AB-9	PG 64-28	2.382	0.0416	11.4	447	13.5	−30.6	15.7	19.32	22.1
AB-10	PG 64-22+SBS	2.050	0.0358	16.5	841	18.6	−34.5	10.4	37.17	16.5

**Table 2 materials-16-07412-t002:** Statistical correlations between spiral tightness (b), spiral cracking energy (*E_Spiral_*), embrittlement temperature (*T_EMB_*), and fracture energy (*G_f_*) for hydrocarbon polymeric materials.

		Correlation			
		*b*	*E_Spiral_*	*T_EMB_*	*G_f_*
Correlation	b	+	-	+	-
	*E_Spiral_*	-	+	-	+
	*T_EMB_*	+	-	+	-
	*G_f_*	-	+	-	+

## Data Availability

Some or all of the data that support the findings of this study are available from the corresponding author upon reasonable request.

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
