# Peer review of "Novel Approach in Fracture Characterization of Soft Adhesive Materials Using Spiral Cracking Patterns"

_materials, 2023, doi:10.3390/ma16237412_

Round 1

Reviewer 1 Report

Comments and Suggestions for Authors

This paper study the "Innovative Method for Analyzing Soft Adhesive Materials' Fracture Characteristics through Spiral Cracking Patterns", which is an interesting topic, while there are still lots of problems that need to be addressed.

1. The abstract does not clearly describe the results of this paper, please revise it.

2. The introduction is not sufficient, there are lots study that study the asphalt binder and mixture low temperatures but not mentioned, for example:

Asphalt binder cracking device test used in this study: https://doi.org/10.1016/j.jclepro.2021.127949

asphalt mixture low-temperature disc-shaped compact tension (DCT) test, shear bond test, and aged asphalt performance are used in these studies:

https://doi.org/10.1016/j.cscm.2023.e01847

https://doi.org/10.1016/j.cscm.2023.e02175

https://www.mdpi.com/2071-1050/14/17/10987

I believe these references can improve the background of the low-temperature performance evaluation part in the introduction part.

3. Figure 2 shows the rigid substrate, could you please show the application of asphalt in the rigid substrate, that has a spiral crack?

4. where are the validation data results of the CNN model? How many samples are used in this CNN model?

5. please show the specification of the test in Figure 6.

6. Figure 7 and Figure 8 are not to be seen, please revise.

7. where is the error bar of the results in Table 1.

8.  Figure 9 (a) and Figure 10 are hard to be seen.

9. The conclusion part needs to be improved, it is not easy to understand now. 

Comments on the Quality of English Language

Moderate editing of the English language required

Reviewer 2 Report

Comments and Suggestions for Authors

This is interesting and timely work. Some comments:

1.       Abstract: it is overall easy to follow, however, it is a bit too general, please provide more technical information.

2.       Introduction: there are two types of fractures in asphalt binders: fatigue cracking and thermal cracking. The cracking mechanism and propagation are completely different. Which do the authors want to address?

3.       Introduction: it is more like a background instead of introduction, and very limited literature review. Please reorganize it.

4.       What is the key contribution in this study?

5.       How many replicators were performed for each material?

6.       Figure 4 is an illustration; can the authors provide a photo for the test?

7.       The overall analysis is reliable and well organized.

8.       In the future study, how do the authors consider the effect of glass transition temperature?

Comments on the Quality of English Language

Very easy to understand.

Reviewer 3 Report

Comments and Suggestions for Authors

Dear Authors,

thank you for your excellent manuscript, here will be minor remarks:

1. "Experience showed that presence of mud cracks in spiral samples are the source of variability (number of mud cracks, their lengths, and their formation locations vary from one sample to another)" Can you please add here or references or addition on the information on projects/objects maybe some photos from field?

2. "Spiral cracks are produced in the lab under a controlled condition" please indicate range of T, etc. in bullet form with possible variations. maybe even the paragraph 120-149 lines, make a bit more structurally visually attractive. 

3. Conclusions in a bullet form will be more attractive for reading.

Comments on the Quality of English Language

Ok

Round 2

Reviewer 1 Report

Comments and Suggestions for Authors

After a review of the paper "Novel Approach in Fracture Characterization of Soft Adhesive Materials using Spiral Cracking Patterns", the current version has updated all the comments and stands as comprehensive and aligned with the requirements of a journal paper.

Comments on the Quality of English Language

Minor editing of English language required